# The Diabetes Remission in India (DiRemI) study: Protocol for a prospective matched-control trial

**Pramod Tripathi[1,2], Nidhi Kadam[1]\*, Diptika Tiwari[1], Thejas Kathrikolly[1], Anagha Vyawahare[1], Baby Sharma[1], Malhar Ganla[2], Banshi Saboo[3]**

**1** Department of Research, Freedom From Diabetes Research Foundation, Pune, Maharashtra, India,
**2** Department of Management & Exercise Science, Freedom from Diabetes Clinic, Pune, Maharashtra, India,
**3** Department of Medicine, Dia Care- Diabetes Care and Hormone Clinic, Diabetology, Ahmedabad, Gujarat, India

\* research@freedomfromdiabetes.org

**Data Availability Statement:** No datasets were generated or analysed during the current study. All relevant data from this study will be made available upon study completion.

## Abstract

### Background

The global rise in diabetes, particularly in India, poses a significant public health challenge, with factors such as limited awareness, financial strain, and cultural considerations hindering its effective management. Although lifestyle changes have shown promising results, their consistent implementation and maintenance continue to pose challenges. Most studies have focused primarily on dietary modifications, overlooking other essential aspects of lifestyle intervention. The DiRemI study aims to address these gaps by evaluating the efficacy of a comprehensive one-year program that combines diet, exercise, psychological support, and medical management to achieve weight loss, diabetes remission, and improved glycemic control among patients with type 2 diabetes (T2D) in India, while also considering the unique needs of the Indian population.

### Methods

The DiRemI study is a prospective, open-label, matched-group trial aimed at assessing the impact of a one-year online integrated intensive lifestyle intervention (ILI) comprising dietary modifications, physical activity, psychological support, and medical management on weight loss and remission in adult T2D patients (aged 30–70 years), with a body mass index (BMI) between 25 and 35 kg/m$^2$, and disease duration of <15 years. ILI will be compared with routine medical care (RMC). Participants will be recruited from three clinics: one providing ILI and two others providing RMC. The co-primary outcome will be weight loss and remission at 12 months, with a follow-up at 18 months. The proposed sample size is 360 participants (180 each in intervention and control group).

### Discussion

The DiRemI study represents the first large-scale remission study in India to show the effectiveness of an integrated approach in the remission and management of T2D and its

**Funding:** The author(s) received no specific funding for this work.

**Competing interests:** The authors have declared that no competing interests exist.

complications. The findings of this study hold the potential to report evidence-based strategies for managing T2D both in India and globally, thus alleviating the substantial burden of diabetes on public health systems.

## Trial registration

Clinical Trials Registry, India (Registered Number: CTRI/2023/06/053885).

## Introduction

The worldwide prevalence of diabetes mellitus is increasing, posing a significant threat to public health. In 2021, the estimated global incidence of diabetes in adults was 537 million, with 90% of cases of type 2 diabetes (T2D) [1, 2]. According to the International Diabetes Federation (IDF) (2019), globally, India has the second-highest number of individuals with diabetes at 77 million and the number is projected to increase to nearly 134 million by 2045 [3]. It further estimates that India contributed the highest to diabetes-associated complication deaths in South Asia in 2019 [3]. Therefore, the overbearing public health burden of T2D in the region necessitates effective management and intervention strategies.

Efficient management of diabetes and its associated complications in India is hindered by several factors, including limited awareness of diabetes, its risk factors, prevention methods, insufficient healthcare infrastructure, economic challenges, and non-adherence to medication. [4]. Patients in India face significant financial strain due to out-of-pocket expenses, particularly in the absence of adequate insurance schemes and policies [5, 6]. In addition, social stigma adds to the mental stress they experience [4]. All of these factors can lead to underdiagnosis and poor treatment-seeking behaviour, leading to an increased risk of complications and comorbidities [7]. India is a country with diverse culture and food habits with studies indicating that this population is unique in dealing with emotional distress, perceived stress, and other psychological conditions related to T2D [8, 9]. Further, the South Asian phenotype and genotype are distinct, as evidenced by central obesity, even at normal BMI, which makes them more prone to diabetes and its complications at relatively low BMIs in both native and immigrant populations [8]. Therefore, there is an urgent need to address the public health concerns surrounding T2D in India.

Pharmacological management can delay, but not entirely prevent, the development of diabetes-related complications. Achieving T2D remission, which is defined as maintaining HbA1C levels below 6.5% for at least three months without the use of glucose-lowering medications, may be the most effective way to reduce the morbidity and mortality associated with the disease [10, 11]. T2D remission has been reported in patients treated with metabolic surgery, intensive pharmacotherapy, and significant lifestyle modifications [12], each with its own merits and limitations. Among them, lifestyle interventions may be the most economical.

Previous studies using intensive lifestyle interventions (ILI) have reported successful T2D remission. However, despite considerable evidence of remission, lifestyle management is not considered the standard first-line of care because of the challenge of making patients adopt and adhere to the lifestyle changes, which includes complete dietary modification, behavioural changes, a specific exercise regime, regular monitoring of blood sugar levels, and staying optimistic and motivated for a longer duration [13–15].

A systematic review by Karera et al. (2023) [16] indicated that most studies employing ILI concentrated on dietary modifications through total diet replacement and medication

cessation in individuals with higher body mass indices (>27 kg/m$^2$) and shorter disease durations. Notably, none of these studies considered the mental health and existing dietary patterns of the study participants, while only one study by Taheri et al. (2018) [17] included exercise as a part of ILI. This highlights the lack of studies using a comprehensive approach incorporating diet, exercise, and psychological support, along with medical management for patients with a longer duration of T2D.

The Diabetes Remission in India (DiRemI) study aims to evaluate the efficacy of the one-year Freedom from Diabetes protocol, which integrates diet, physical activity, psychological support, and medical management, compared to standard medical care to achieve weight loss, diabetes remission, and improved glycemic control in patients with T2D in the Indian population. The program is individualized & designed considering the cultural barriers, food choices, physical activity challenges, and psychological aspects of individuals with T2D in India.

### Specific objectives (as specified during ethics submission)

The primary objective of the intervention is to achieve a weight reduction of ≥ 10% and an HbA1c level of <6.5% without pharmacotherapy, with the ultimate goal of achieving diabetes remission. The secondary objective of the intervention is to evaluate its effect on other aspects of diabetes outcomes, including mental health (anxiety and depression), comorbidities (hypertension and dyslipidaemia), diabetes-related complications (nephropathy), and biochemical parameters (serum lipids, liver function tests, and kidney function tests).

## Methods
### Design

The present study is a prospective, open-label, matched control trial. Ethical approval for the study was granted by the Institutional Ethics Committee (IEC) (Ref. no. FFDRF/IEC/2023/3). The trial is prospectively registered with the Clinical Trials Registry-India (CTRI) (Ref. No. CTRI/2023/06/053885) (S1 File). This study will adhere to the standards for clinical research outlined in the Declaration of Helsinki. The protocol is designed according to the SPIRIT guidelines (S1 Checklist).

### Setting and eligibility criteria

The proposed study will be conducted at the Freedom from Diabetes Clinic, which operates on an online subscription-based model of 1-year duration for the management of diabetes (intervention group). Additionally, the study will include participants from two private clinics in Pune and Ahmedabad (India) (Control Group), which provide standard medical care based on established guidelines for individuals with diabetes. All individuals with T2D who enrol in these centres between March and August 2024 and meet the eligibility criteria will be recruited for this study Table 1.

### Sample size determination

We calculated the sample size for our study using Bonferroni correction to adjust for two primary outcome measures: weight and HbA1c. With an overall alpha level of 0.05, the adjusted significance level for each outcome was set to 0.025. Assuming a power of 95%, the required sample sizes were determined based on Cohen's d effect sizes of 0.445 for weight and 0.635 for HbA1c, based on a previous study by Lean et al. (2018) [18]. The calculations indicated a sample size of 132 participants per group for weight and 65 participants per group for HbA1c level. To ensure adequate power for both outcomes, we selected the larger sample size

**Table 1. Eligibility criteria for the DiRemI study.**

| Inclusion Criteria | Exclusion Criteria |
|---|---|
| Age between 30 and 70 years | Pancreatitis or drug-induced diabetes (Steroids) |
| Confirmed diagnosis of type 2 diabetes, which may be managed with oral hypoglycemic agents, insulin, Ayurvedic antidiabetic medications, or have an HbA1c level of $\geq$ 48 mmol/mol (6.5%) with or without medication | Other types of diabetes (type 1 diabetes mellitus, diabetes insipidus, maturity-onset diabetes of the young, latent autoimmune diabetes in adults, or Gestational Diabetes) |
| Body mass index (BMI) between 25 kg/m$^2$ and 35 kg/m$^2$ | Advanced complications related to nephropathy (eGFR<30 mL/min/1.73 m$^2$, urine microalbumin >1000 mg), retinopathy (severe non-proliferative diabetic retinopathy and proliferative diabetic retinopathy), neuropathy (peripheral artery disease or diabetic neuropathy resulting in complete numbness and loss of sensation, amputations, ulcers, foot deformities, Charcot's foot) |
| Duration of type 2 diabetes less than 15 years | History of heart attack, cardiac arrhythmia, bypass or stent placement surgery, ischemic heart disease, angina (class II or III), myocardial infarction within the previous 6 months, reported ejection fraction <40%, abnormal ECG, and positive stress test results (moderate or severe) |
| Ready to provide written informed consent | Known history of cancer |
| Agree to undergo the examination, lifestyle intervention program, and post-program evaluation | Pregnant and lactating women |
| Well-oriented in time, space, and as a person | Patients who have required hospitalization for any diabetes-related complications in the last 6 months or for depression or are on antipsychotic drugs |

requirement, resulting in a total sample size of 132 participants per group. To account for the potential dropout and loss to follow-up of 30%, the final sample size was estimated at 360 (180 in each group).

## Sampling and recruitment process

All patients participating in the program (intervention group) or attending the two clinics (control group) will be screened for eligibility criteria. A list of all eligible participants from each of the three clinics will be compiled every month. Participants will be selected for the study based on computer-generated random numbers until the desired sample size is achieved.

Prior to participation, all participants will be required to provide written informed consent. The consent form will outline the purpose, procedures, benefits, risks, participation requirements, contact information for the Principal Investigator, and information regarding withdrawal from the study. Consent will be collected by a trained member of the research team and stored electronically. Data will be collected at four time points: at the start of the study, at six months, 12, and 18 months after the commencement of the study.

The study period, including the schedule of enrolment, intervention, and assessment of the study groups, is shown in Fig 1.

## Study procedures

Standardised questionnaires will be used at all three centres to ensure uniformity in the data collection. Additionally, translations and assistance in completing the questionnaires in the local languages (Hindi, Gujarati, and Marathi) will be provided to those who require them by a trained member of the research team.

| STUDY PERIOD | | | | | | |
|---|---|---|---|---|---|---|
| **TIMEPOINT** | T1 | T2 | | T3 | | T4 |
| **MONTHS** | 0 BASELINE | 3 | 6 | 9 | 12 | 18 FOLLOW-UP |
| **ENROLLMENT:** | | | | | | |
| Eligibility | X | | | | | |
| Informed consent | X | | | | | |
| Allocation | X | | | | | |
| **INTERVENTIONS:** | | | | | | |
| Routine Medical care | X | X | X | X | X | |
| Lifestyle intervention | X | X | X | X | X | |
| **ASSESSMENT:** | | | | | | |
| Baseline data collection | X | | | | | |
| Intermediate data collection | | | X | | | |
| End-line data collection | | | | | X | |
| Follow up on the status | | | | | | X |
| Analysis and report writing | | | | | X | X |

**Fig 1. SPIRIT timeline: Schedule of enrolment, intervention, and assessment for both study groups at four different time points T1 -T4.**

**Anthropometric and socio-demographic.** During baseline assessment, participants will be requested to provide self-reported data on sociodemographic variables, including age, gender, education, and marital status, as well as anthropometric measurements such as height and weight. Detailed instructions on the precautions to be taken while measuring weight will be provided to ensure the accuracy of the self-reported data (S1 Appendix). The data will be reported as the average of three readings. Body mass index (BMI) will be calculated.

**Biochemical.** Data on the following biochemical tests will be collected from lab reports done at Government certified- NABL (National Accreditation Board for Testing and Calibration Laboratories, Govt. of India) labs; Primary: HbA1c, fasting blood sugar (FBS), postprandial blood sugar; secondary: lipid profile (total cholesterol, HDL, LDL, and VLDL), thyroid profile (thyroid-stimulating hormone), fasting insulin, serum creatinine levels, eGFR, urine microalbumin, liver function parameters, high sensitivity C-reactive protein (hs-CRP), C-peptide, Vitamin D, and Vitamin B12 will be collected based on availability in the submitted reports. Homeostatic model assessment of insulin resistance (HOMA-IR) and homeostatic model assessment of beta cell function (HOMA-B) will be calculated using standard formulas to assess insulin resistance and beta-cell function, respectively [19].

**Mental health assessment.** The validated Patient Health Questionnaire (PHQ-9) and Generalised Anxiety Disorder (GAD-7) scale will be used to assess the prevalence of depression and anxiety, respectively.

The PHQ-9 is a 9-item self-report screening tool for depression. Each item is rated on a 4-point scale from 0 to 3 (0 = never, 1 = several days, 2 = more than half the time, and 3 = nearly every day). The overall scores range from 0 to 27, with higher scores indicating greater symptom severity. Scores 5, 10, 15, and 20 represent cut-offs for mild, moderate, moderately severe, and severe depression, respectively. The PHQ-9 has a sensitivity of 88% and a specificity of 88% for detecting depression [20].

The GAD-7 is a 7-item self-report screening tool for anxiety. Each item is rated on a 4-point scale from 0 to 3 (0 = never, 1 = several days, 2 = more than half the time, and 3 = nearly every day). The overall scores range from 0 to 21, with higher scores indicating greater symptom severity. Scores 5, 10, and 15 represent cut-offs for mild, moderate, and severe anxiety, respectively. The GAD-7 has a sensitivity of 89% and a specificity of 82% for detecting anxiety [21].

| Interventions | Diet Protocol | Exercise Protocol | Psychological Protocol | Medical Protocol |
|---|---|---|---|---|
| Phase 1 | Adjustment to plant-based diet. Average calorie intake 1200-1400 kcal/d | To promote muscle activation and blood circulation for post-meal glycaemic control, with a special focus on gaining strength, flexibility and stamina, and improving posture and breathing. | Understanding one's stress and anxiety levels, Individual counselling based on the presence of anxiety and/or depression. | All the 4 phases of intervention will follow medical management which includes analysing medical reports at monthly/ 3 monthly /6 monthly intervals, generating medical and supplement prescriptions, daily monitoring of blood sugar levels through mobile application and adjusting insulin/ medicine dosage and tracking physical parameters related to weight and BP; thus, closely monitoring participants. |
| Phase 2 | Alkalizing diet through Intermittent and Juice fasting. Calorie intake ranging from 300-1000 kcal/d | Customized 3-2-1 exercise plan based on BMI, age, associated health conditions, and personal preferences. | Journaling and meditation practices for stress release for 56 days | |
| Phase 3 | Customized diet plan for further weight loss with an advanced Intermittent Fasting Protocol or shifting to muscle gain diet if weight target is achieved. Average calorie intake 1400-1600 kcal/d. | Incorporate selected athletic specialization in core activities like weight training (gym), running, cycling, swimming, yoga, trekking, etc in a 3-2-1 pattern for long-term sustainability. | Creating detailed individual special health goals and affirmations. Convert these goals into Vision Boards. Practices for inculcating these goals & vision boards into one's subconscious mind by applying Law of Attraction. [27] | |
| Phase 4 | Maintenance and advanced muscle-building diet. Reintroduction of earlier restricted food items. Average calorie intake 1600-1800 kcal/d | Establish a consistent level of fitness through a set routine of 3-2-1 exercises and promote independence by educating about methods of progression and calendarizing them over a year. | | |

**Fig 2. DiRemI intervention.**

## Intervention

The Freedom from Diabetes protocol is a comprehensive one-year ILI focused on four key components: dietary modifications, physical activity, psychological support, and medical management. Each participant is supported by a dedicated team of experts, which includes a physician, a dietitian, a physical therapist/certified yoga professional, a psychologist, a mentor (who is a past participant volunteering to guide the new participant), and a monitor (for follow-ups and reminders for lab tests and appointments). Participants also have access to a mobile application that facilitates communication with the team via voice/video calls and text messages.

The interventions implemented in the DIRemI study are shown in Fig 2. The four protocols are divided into several phases, each designed to introduce incremental modifications, initially focusing on diet and exercise. Subsequently, both are enhanced by the addition of emotional stress release and goal setting. The ultimate objective of the phases is to render the participants self-sufficient and able to sustain the changes. Progression to the subsequent phase is contingent on achieving the predetermined target for the previous phase. The goals for each phase of the program are established collaboratively with the patient's healthcare team, which comprises a dedicated physician, dietitian, and physical therapist. Throughout the program, these objectives serve as a guide for patient progress and are periodically reviewed and adjusted as necessary.

**Dietary modifications.** Dietary intervention is focused on a plant-based diet, individualised based on the BMI and associated medical conditions of the participants. The four distinct phases of dietary intervention are adjustment, accelerated weight loss, muscle building, and maintenance. As part of the dietary intervention, participants are primarily introduced to consuming greens, lentil-based food items, sprouts, and vegetable salads. Soaked nuts such as walnuts, almonds, seeds, and fruits are also recommended for consumption, based on individual blood sugar levels. Sample diet charts are included in the supplementary file (S2 Appendix).

Phases 1 and 2 primarily focus on weight loss through adjustment to a plant-based alkaline diet which helps reduce inflammation. Consumption of milk and milk products, meat, eggs, and poultry are restricted during these phases. In Phase 1, the focus is on a shift from the usual high-carbohydrate unbalanced diet to a balanced diet considering all macro-and micronutrients by increasing the consumption of cooked and raw vegetables along with green smoothies, providing a rich variety of antioxidants and phytonutrients. The reported average calorie

| Week | Mon | Tues | Wed | Thurs | Fri | Sat | Sun |
|------|-----|------|-----|-------|-----|-----|-----|
| | | | | Number of Meals/Day | | | |
| 1st | 3 | 2 | 3 | 2 | 3 | 2 | JF |
| 2nd | 2 | 2 | 2 | 2 | 2 | 2 | JF |
| 3rd | 2 | 1 | 2 | 1 | 2 | 1 | JF |
| 4th | 1 | 1 | 1 | 1 | 1 | 1 | JF |
| 5th | 1(D) | 0 | 1(D) | 0 | 1(D) | 0 | JF |

**Fig 3. Intermittent fasting protocol: 3, all three meals; 2, Brunch/Lunch and Dinner; 1, Brunch/Lunch or Dinner; 1(D), Dinner only; JF, Juice fasting.**

intake for Indian T2D patients is 1547 kcal (95% CI 1486 to 1608) and constitutes 64% of the total energy intake [22]. The average calorie intake for Phase 1 is reduced to 1200–1400 kcal/ day. In the second phase, one month of intermittent fasting and juice fasting is recommended to aid in faster weight loss and detoxification [23], depending on the BMI of the participant; those already achieving a normal BMI ($<23$ kg/m$^2$) do not undergo intermittent fasting but only juice fasting. The intermittent fasting and juice fasting protocols are described in Fig 3. The average calorie intake during this period ranges from 300–500 kcal on days of juice fasting to 400–500 kcal on single-meal days; the average intake is 800–1000 kcal for days when two meals are consumed. The recipes for juices used in juice fasting (red, white, and green) have been previously described [24]. During periods of fasting, the participants are instructed to closely monitor and report their sugar levels to their assigned physician through a mobile application and accordingly, drug-dose adjustments are done.

Once the target BMI (based on the initial goal setting-23-25 kg/m$^2$ or a 3–5 point drop) is achieved, calorie intake is gradually increased to 1400–1600 kcal to facilitate muscle building in Phase 3. In addition, protein intake is increased to 75–80 grams depending on the intensity of the recommended exercises. The focus here is on building up muscle mass and increasing stamina. The last phase emphasises on maintenance diet. In this phase, once the blood sugar levels have stabilised, other food items restricted from the regular diet are reintroduced phase-wise under the close supervision of the dietitian. This helps participants to understand the portion and frequency control required to sustain the results. This phase also helps participants maintain good glycemic control despite the reintroduction of restricted foods. Based on the chosen long-term exercise plan, calorie intake may be increased to 1600–1800 kcal/d (for those who no longer need to lose further weight). This phase is designed to enable individuals to effectively manage their diet and sustain an exercise plan for better glycemic control, while maintaining weight loss post-intervention without support.

During each phase, the blood sugar levels are closely monitored.

**Physical activity modifications.** The phase-wise physical activity protocol is shown in Fig 2. The exercises primarily focus on building and sustaining strength, flexibility, and stamina in the long run by adopting a 3-2-1 pattern per week (three days for strength, two days for flexibility, and one day for stamina exercises) over one year. The exercise duration is 30–45 minutes per day, 6 days a week. Sample exercise plans are included in the supplementary file (S2 Appendix).

Phase 1 exercises aim to promote blood circulation, muscle activation, and anti-gravity exercises to enhance post-meal glycemic control [25]. Additionally, exercises focused on improving posture, breathing, flexibility (yoga), and strength are recommended. Based on the

progress made during Phase 1, the activities for Phase 2 are recommended from the following five categories: muscle gain, weight loss, core strengthening, yoga, and specialised exercises for senior participants. This phase focuses on strengthening the upper and lower limb muscles and improving flexibility, cardiopulmonary endurance, balance, and coordination in addition to weight loss. The third phase of the intervention focuses on individualised athletic specialisation based on age, BMI, and preference by increasing the intensity and duration of the selected exercise form. Solo exercises such as weight training, swimming, running, cycling, and yoga, are suggested. To establish a consistent state of fitness and sustenance after the completion of the program, the final phase focuses on periodized exercise plans for managing various goals and activities, enabling the participant to independently create and design a personal workout plan for the following week, month, and year.

**Psychological support.**   The third protocol of psychological intervention is designed to address stress and anxiety levels by raising awareness of the mind-body connection. At baseline, all participants will be assessed for depression and anxiety using validated tools: the Participant Health Questionnaire (PHQ) and Generalised Anxiety Disorder (GAD) scale. Those who exhibit moderate to severe anxiety and/or depression are referred to in-house psychologists for one-on-one counseling sessions tailored to their individual needs. Drawing from a range of therapeutic techniques, including Cognitive Behaviour Therapy (CBT), Rational Emotive Behaviour Therapy (REBT), Neuro-Linguistic Programming (NLP), Clinical Hypnotherapy, Life Coaching, Pranic Healing, and Eclectic Therapy, therapists work to meet the unique needs of each participant.

All participants, regardless of their anxiety and depression scores, attend 56 days of group therapy sessions during the one year; these incorporate guided meditation and journaling. The intervention is centred on cultivating positive energy through principles such as the law of attraction, gratitude, and affirmation [26]. Engaging in cognitive activities, such as creating a vision board, serves as a proactive measure for achieving personal goals and enhancing emotional intelligence [26, 27]. The intervention also emphasises the importance of long-term health management and maintenance through online group sessions following the program.

At the end of 6 months, the PHQ and GAD are re-administered to assess the improvement in the mental health outcomes.

**Medical management.**   The primary focus of medical management is to closely monitor participants during the intervention. The assigned physician is accountable for evaluating the participant's health condition at the outset by considering a comprehensive medical history of diabetes, its potential complications, and any co-existing conditions, along with biochemical analysis of the parameters listed previously. Following the correction of micronutrient deficiencies, if any, the physician adjusts the medication dosage according to the participant's weight, BMI, and daily self-reported blood sugar level (BSL) through the mobile application. Once blood sugar levels are stabilised, participants are asked to report them less frequently (weekly/fortnightly). The physician considers the participant's history of comorbidities for further management and maintains strict control over the HbA1c levels throughout the year. In addition to daily dosage modifications, the physician is required to conduct at least three medical consultations with the participant throughout the one-year program.

**Mode of delivery.**   The intervention is delivered online through video meetings and conferences. Throughout the year, the team of experts keep in touch with the participants through an average of 63 calls per year. In addition to the 12 monthly sessions, group sessions covering the primary focus for that month, additional group activities such as mid-month catch-up sessions, and diet/exercise-wiser sessions are conducted to check and improve adherence and motivation. Participants have access to a dedicated mobile application to communicate with their team and regularly track and update their vitals (fasting and postprandial blood sugar

levels, weight, and blood pressure) along with their diet and exercise regimen. Participants are also provided with region-specific plant-based recipes to avoid drastic changes in their existing diet.

For exercise recommendations, fitness level, age, physical limitations, and preferences are considered. Phases 1 and 2 have pre-recorded exercise sessions which are relayed in the morning and evening for six months, catering to differing needs. Additionally, other exercise videos for different physical limitations like spondylosis, frozen shoulder, and knee issues that are provided based on the requirement and recommendation. The participants are guided individually by their assigned physical therapist.

Overall, diet, physical activity, and stress management are tailored to each participant's individual needs, considering their comorbidities, age, sex, and fitness levels. In total, there are 600+ exercise videos, session videos, meditation audios, PDFs, and 400+ recipes to choose from, giving the participants variety and progression.

**Adherence check.** Participants in the program are regularly encouraged to follow the protocol, and their adherence is assessed through monthly checks using a mobile application. These checks involve short multiple-choice questionnaires designed to evaluate participants' adherence to the relevant interventions for that period. The monthly adherence scores are calculated and used to provide feedback to experts, who can then take appropriate action. At the end of the 12-month intervention period, monthly scores are combined to generate an overall adherence score for each participant. The purpose of these measures is to ensure that participants adhere to the program's protocol and provide feedback to experts for any necessary actions.

## Control group (Routine medical care)

The control group will receive routine medical care along with education for management of T2D and its associated comorbidities based on current clinical guidelines.

## Outcome measures

The primary outcome measures are a) Weight loss of more than or equal to 10% and b) Remission of T2D (HbA1c less than 6.5% for at least 3 months without medication). Secondary outcomes include a) reduction in dosage and/ or frequency/ class of drug or complete stoppage of diabetes medication, b) improved glycemic control (HbA1c less than 7% with or without medication), c) reduction in fasting glucose levels (below 126 mg/dl) and PP blood glucose levels (below 200 mg/dl) with or without glucose-lowering medications), d) improvement in diabetes-related complications (nephropathy- improvement in terms of eGFR>90 mL/min/ 1.73m$^2$), e) improvement in comorbidities (hypertension- blood pressure ≤140/90 and dyslipidaemia- Total cholesterol <200 mg/dl, triglycerides <150 mg/dl, and HDL >40 mg/dl for males and >50 mg/dl for females), and f) improvement in mental health outcomes (anxiety and depression- Reduction in the score—below 10).

## Adverse events record, withdrawal, and safety measures

Participants will be regularly monitored for any adverse events during the intervention by the assigned physician by reporting on the mobile application. The communication between the participant and the physician is daily to monitor and adjust the drug dosage until the blood sugar levels have stabilised. Any adverse events that occur during the intervention will be documented along with the methods employed to address them. Any adverse effects that may lead to a participant's discontinuation or occur during the course of the study will be followed up

until a satisfactory solution is reached. This information will also be included in the final report.

Participants will have the right to withdraw from the trial at any time without providing any explanation. The study will discontinue the subject's participation if any of the following conditions occur: a) the subject fails to comply with the study protocol and requirements; b) there is a significant deviation from the study protocol; c) the subject becomes pregnant during the study; d) the subject develops a new illness that warrants exclusion based on the criteria mentioned above; e) continuation in the study is deemed harmful to the subject's health, f) the subject is lost to follow-up, and g) the trial is cancelled.

Any deviation from this protocol will be reported to the Ethics Committee, and approval will be sought.

## Data management, processing, and dissemination

At the outset of the baseline study, each participant will be assigned a unique identifier that will remain confidential throughout the study. All collected data will be securely stored in the data management software, and access to this information will be restricted without prior authorisation. The members of IEC and CTRI will monitor all data. De-identified/coded data will be extracted from the data management system as CSV files at all three centres. This will then be transferred to Microsoft Excel and subsequently to SPSS (version 21.0; IBM Corp., Armonk, NY, USA). In compliance with the ICMR (Indian Council of Medical Research) Ethical Guidelines, the data will be stored for a minimum of 5 years. These data may be used for follow-up of participants to assess the long-term effects of the intervention.

The results will be published in open-access journals and presented at National & International conferences. All findings from the trial will be notified to the IEC and CTRI.

## Statistical analyses

Primary outcome measures will be analysed using two-sided statistical tests with a type I error rate of $\alpha = 0.05$. Primary analysis will be performed using the intention-to-treat method. Categorical data will be compared between the two groups using chi-square or Fisher's exact tests. Continuous data will be evaluated using parametric or non-parametric tests based on data distribution.

For outliers (± 3 standard deviation), efforts will be made to cross-check and obtain the correct values. Failing this, the case will be considered as missing data. In the case of random missing values, two separate analyses will be performed: one including the cases with missing data by choosing pairwise deletion and the other excluding it using listwise or case-wise deletion. A complete case analysis will be considered if differences are observed between the two analyses. In the case of non-random missing values, the imputation method may be used by replacing the missing value with the mean for that variable.

Potential adjustments for baseline values, age, gender, and disease duration may be considered. Multivariable linear and logistic regression analyses will be used to determine predictors of remission. Finally, subgroup analyses may be performed based on factors such as gender, age, comorbidity, diagnosis length, and severity to investigate potential variations in treatment responses.

## Trial status

The recruitment process commenced on 1 March 2024 and is expected to finish by 31 August 2024.

## Discussion

Current diabetes management practices typically involve the use of medications to control blood sugar, lipids, and blood pressure [28]. However, these pharmacological interventions do not necessarily address the long-term complications associated with type 2 diabetes (T2D), particularly in cases of non-compliance with treatment or ineffective treatment regimens [29]. A more comprehensive approach to diabetes management, including dietary modification, regular physical activity, and psychological support, may help prevent long-term complications and slow disease progression.

Our study, which spans a period of one year, seeks to effectively manage and achieve remission of T2D through the implementation of dietary interventions, exercise regimens, psychological therapies, and medical management. The DiRemI study represents the first trial in India aimed at T2D remission using an online delivery mode, with a focus on regional, biological, cultural, and lifestyle differences between the study participants. The innovative aspect of our study is the integration of a personalised, holistic, and sustainable healthcare regime that draws on the traditional aspects of the Indian way of life to enhance adherence to treatment protocols. Our study will compare the outcomes of personalised holistic interventions, including diet, exercise, psychological therapies, and medical management, with those of a control group receiving routine medical care.

Several studies have demonstrated the effectiveness of low or very low carbohydrate, low fat, intermittent energy restriction, total diet replacement, and Mediterranean diet for weight loss in participants with T2D [18, 30–36]. However, our dietary intervention program differs from these studies in that it focuses on modifying existing diets by adjusting calorie intake to align with an individual's health needs, preferences, and dietary norms. This approach ensures that positive dietary changes are sustainable beyond the program's immediate timeframe and extend into the post-intervention period. In our trial, in conjunction with dietary modifications, individuals will receive tailored guidance to enhance stability, flexibility, and endurance across a range of physical activities and yoga, based on their age, body weight, and fitness levels, as part of the sustainable remission objective.

The importance of mental health is unparalleled and closely connected to effective T2D management. In addition to dietary and exercise adjustments, this trial encompasses a multifaceted approach to both physical and mental well-being, which has not been extensively addressed in previous studies. Research has demonstrated that participants with T2D who experience depression are twice as likely to exhibit non-adherence to medications for diabetes, blood pressure, and cholesterol when compared to non-depressed participants [37]. Moreover, individuals with poorer diabetes self-care tendencies generally exhibit heightened levels of psychological distress, diminished social support, and decreased self-efficacy [38]. In the proposed trial, T2D participants will receive psychological support from trained psychologists to help address and cope with life stressors and achieve optimal disease management.

Another unique feature of our study is the focus on medical management. Unlike previous studies on ILI [17, 39] that completely withdrew all diabetes medicines before the commencement of the trials, in our study, the physician will ensure systematic tapering of medications based on daily reported blood sugar levels. This approach will ensure appropriate medical management of participants with a longer duration of diabetes and, hence, more complications. Furthermore, medical management will address and correct nutritional deficiencies through supplementation.

Our study differs from many other studies on ILI, which have generally focused on specific population demographics (e.g. younger age and newly diagnosed T2D) [35, 39, 40] and limited interventions targeting specific features, such as low/very low-calorie diets, diet replacement,

and weight loss, with less emphasis on physical activity and intensive exercise training [36, 41]. Our study represents a unique approach that incorporates a comprehensive, personalised methodology specifically designed to accommodate the distinct features of each participant, with a particular focus on a demographically varied group of individuals aged between 30 and 70 years, who have been living with diabetes for up to 15 years.

In this context, this study aims to contribute to the growing body of evidence supporting the effectiveness of holistic interventions in managing T2D, particularly in diverse populations, such as those in India. By addressing not only physical health, but also mental well-being and cultural considerations, our findings may pave the way for more tailored and effective diabetes management strategies globally.

Our study has a few limitations that should be considered. First, the socioeconomic status of the participants may limit the generalisability of our findings to the broader Indian context because our online diabetes management program operates on a subscription-based model that may restrict access to those who can afford it. Future trials adopting a randomised-controlled design may be planned based on the outcome of the present study to address this. Second, the study relies on self-reported parameters, such as anthropometry, which may be subject to data errors despite detailed instructions and regular monitoring of incoming data. Additionally, although our study is not a true randomised controlled trial, we have chosen a matched-control group for our intervention group which is more acceptable in a real-world setting. Finally, due to the comprehensive nature of our planned intervention, we will not be able to examine the impact of individual components. Notwithstanding these limitations, we believe that our research will provide valuable insights into the effectiveness of our holistic diabetes management program.

## Conclusion

Our protocol of diabetes management proposes an alteration from a medical treatment-focused approach to a comprehensive lifestyle strategy that incorporates physical activity, psychological support, medical management, and dietary modifications. A crucial aspect of our holistic approach is to make the intervention more practical, customised, and adaptable for participants.

## Supporting information

**S1 File. Clinical trial registration.**
(PDF)

**S1 Checklist. SPIRIT 2013 checklist: A study protocol.**
(DOC)

**S1 Appendix. Weight measurement instruction.**
(PDF)

**S2 Appendix. Sample diet and exercise chart.**
(PDF)

## Acknowledgments

The authors extend their sincere appreciation to Ms. Tanmayi Naik, Ms. Gandhali Bhagwat, Dr Amruta Prabhu, Ms Nupur Akotkar, and Dr Anahita Thomas for their invaluable input in shaping the protocol of this study. We would also like to acknowledge the helpful suggestions provided by Dr. Venugopal V and Dr. Mahesh Kumar Kuppusamy for outlining the

manuscript. The authors also acknowledge the use of *Paperpal* software in improving the language quality of the manuscript. Lastly, we would like to thank all prospective patients who will be part of the study.

## Author Contributions

**Conceptualization:** Pramod Tripathi, Nidhi Kadam.

**Methodology:** Pramod Tripathi, Nidhi Kadam, Banshi Saboo.

**Supervision:** Pramod Tripathi, Nidhi Kadam, Banshi Saboo.

**Writing – original draft:** Diptika Tiwari, Thejas Kathrikolly, Anagha Vyawahare, Baby Sharma.

**Writing – review & editing:** Nidhi Kadam, Malhar Ganla.

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
