## [Decision Letter · Decision Letter 0]

21 May 2024

PONE-D-24-12706The Diabetes Remission in India (DiRemI) Study: Protocol for a Prospective Matched-Control TrialPLOS ONE

Dear Dr. Kadam,

Thank you for submitting your manuscript to PLOS ONE. After careful consideration, we feel that it has merit but does not fully meet PLOS ONE’s publication criteria as it currently stands. Therefore, we invite you to submit a revised version of the manuscript that addresses the points raised during the review process.

The reviewers have raised some concerns regarding the design and analysis of the study. Please address the comments and resubmit the article.

We look forward to receiving your revised manuscript.

Kind regards,

Patricia Khashayar

Academic Editor

PLOS ONE

Journal Requirements:

Reviewers' comments:

Reviewer's Responses to Questions

**Comments to the Author**

1. Does the manuscript provide a valid rationale for the proposed study, with clearly identified and justified research questions?

Reviewer #1: Yes

Reviewer #2: Yes

2. Is the protocol technically sound and planned in a manner that will lead to a meaningful outcome and allow testing the stated hypotheses?

Reviewer #1: Yes

Reviewer #2: Yes

3. Is the methodology feasible and described in sufficient detail to allow the work to be replicable?

Reviewer #1: Yes

Reviewer #2: Yes

4. Have the authors described where all data underlying the findings will be made available when the study is complete?

Reviewer #1: Yes

Reviewer #2: No

5. Is the manuscript presented in an intelligible fashion and written in standard English?

Reviewer #1: Yes

Reviewer #2: Yes

6. Review Comments to the Author

You may also provide optional suggestions and comments to authors that they might find helpful in planning their study.

Reviewer #1: Line 178 was a two sample t-test used to calculate the sample size?

Clarify primary and secondary endpoints. For multiple endpoints, better adjust p values.

How are outliers defined?

Line 427, how can you perform analyses with inclusion of missing data? In linear regression, records with missing will be automatically excluded.

How will non-random missing be handled.

Reviewer #2: Dear Authors,

I am pleased to review your submitted manuscript which attempts to demonstrate the weight loss effects of a comprehensive treatment in patients with type 2 diabetes. The results are intriguing and could undoubtedly be clinically useful. However, there appear to be some significant concerns.

1. The lower BMI limit for study entry is 25. If subjects in this range are enrolled, achieving significant weight loss could also lead to excessive thinness, which cannot be completely ruled out. What are your thoughts on this?

2. Because the intervention is comprehensive, interpreting the results may be challenging even if the outcome measures show a significant difference. Please consider including refinements that allow for specific discussion of the results obtained.

7. PLOS authors have the option to publish the peer review history of their article (what does this mean?). If published, this will include your full peer review and any attached files.

Reviewer #1: No

Reviewer #2: No

---

## [Author Response · Author response to Decision Letter 0]

31 May 2024

RESPONSE TO REVIEWERS

We sincerely thank the editor and reviewers for their valuable and insightful comments, which helped improve the quality of our manuscript. Please find below the detailed response to the reviewers’ comments for your kind perusal and necessary consideration. 

Reviewer Comments

Reviewer #1:

Comments: Line 178 was a two sample t-test used to calculate the sample size?

Response: Thank you for your comment. Yes. A two-sample t-test was used to calculate the sample size based on the difference between two independent sample means, as reported in a previous study by Lean et al. (Ref. 18).

Clarify primary and secondary endpoints. 

Response: Thank you for your comment. We have added details on both the primary and secondary outcomes in the ‘Outcome Measures’ section. Page 18, Lines. 386-396

For multiple endpoints, better adjust p values.

We acknowledge your suggestion in this regard. We have recalculated the sample size for both weight and HbA1c with adjusted p-values. The same has been added to the manuscript as below- in Page 9, Lines 175-184.

“We calculated the sample size for our study using Bonferroni correction to adjust for two primary outcome measures: weight and HbA1c. With an overall alpha level of 0.05, the adjusted significance level for each outcome was set to 0.025. Assuming a power of 95%, the required sample sizes were determined based on Cohen's d effect sizes of 0.445 for weight and 0.635 for HbA1c, based on a previous study by Lean et al. (2018) [18]. The calculations indicated a sample size of 132 participants per group for weight and 65 participants per group for HbA1c level. To ensure adequate power for both outcomes, we selected the larger sample size requirement, resulting in a total sample size of 132 participants per group. To account for the potential dropout and loss to follow-up of 30%, the final sample size was estimated at 360 (180 in each group).”

How are outliers defined?`

Response: Thank you for your comment. All values greater than +/- 3 times the standard deviation will be considered outliers. We have added the same to the ‘Statistical Analyses’ section as below- in Page 20, Line 433.

“For outliers (± 3 standard deviation), efforts will be made to cross-check and obtain the correct values.”

Line 427, how can you perform analyses with inclusion of missing data? In linear regression, records with missing will be automatically excluded.

Response: Thank you for your comment. We agree that in linear or any other regression analysis, records with missing data will automatically be excluded. However, we would like to clarify that, by including missing values, we mean not deleting the entire case for analysis, where other parameters for the case are available. We have clarified this in the text under the Statistical Analyses section as below in Page 20, Line 435-436.

“ In the case of random missing values, two separate analyses will be performed: one including the cases with missing data by choosing pairwise deletion and the other excluding it using listwise or case-wise deletion.”

How will non-random missing be handled. 

Response: Thank you for your comment. Using the data imputation method, we will replace the missing value with the mean of that variable for non-random missing values. We have added the same to the ‘Statistical Analyses’ section as below- in Page 20, Line. 438-439

“In case of non-random missing values, the imputation method may be used by replacing the missing value with the mean for that variable.”

Reviewer #2: Dear Authors, I am pleased to review your submitted manuscript which attempts to demonstrate the weight loss effects of a comprehensive treatment in patients with type 2 diabetes. The results are intriguing and could undoubtedly be clinically useful. However, there appear to be some significant concerns.

Comments 1. The lower BMI limit for study entry is 25. If subjects in this range are enrolled, achieving significant weight loss could also lead to excessive thinness, which cannot be completely ruled out. What are your thoughts on this?

Response: Thank you for your comment. We acknowledge the significance of ensuring that participants did not undergo extreme weight loss. Based on the Asian WHO cutoffs, a BMI of 25 or higher is categorized as obese and 23-24.9 as overweight (WHO Expert Consultation (2004). Lancet (London, England), 363(9403), 157–163. https://doi.org/10.1016/S0140-6736(03)15268-3.; Mishra et al. 2009 https://pubmed.ncbi.nlm.nih.gov/19582986/). Consequently, in our study, we included participants who met the obesity criteria of BMI ≥25 kg/m2.

To support this, the "Personal Fat Threshold" (PFT) hypothesis proposed by Dr. Roy Taylor suggests that all individuals have a specific fat threshold, and losing weight below this threshold can help regain normal glycemic control regardless of BMI. The ReTUNE trial also supports this finding, showing improvements in HbA1c levels, even in those with normal or near-normal BMI with weight loss. (Taylor, R., & Holman, R. R. (2015). https://doi.org/10.1042/CS20140553(Clinical science (London, England: 1979), 137(16), 1333–1346. https://doi.org/10.1042/CS20230586).

Additionally, since the intervention is personalized, the assigned nutritionist and physician will closely monitor the participants during the weight loss phase to prevent any adverse events due to weight loss. 

Comments 2. Because the intervention is comprehensive, interpreting the results may be challenging even if the outcome measures show a significant difference. Please consider including refinements that allow for specific discussion of the results obtained.

Response: Thank you for your comment. The intervention combines four main aspects, as mentioned in the protocol: diet, exercise, psychological support, and medical management. Our hypothesis for the current study is that a holistic approach to managing diabetes and its related comorbidities would be more beneficial than isolating these protocols. We agree that their individual effects are difficult to isolate. We have also acknowledged the same in the limitations section of the Discussion section. However, previous studies combining diet and exercise have also reported a combined effect of the interventions. Similarly, we will also report the effects of holistic interventions rather than isolating them.

---

## [Decision Letter · Decision Letter 1]

17 Jun 2024

The Diabetes Remission in India (DiRemI) study: protocol for a prospective matched-control trial

PONE-D-24-12706R1

Dear Dr. Kadam,

We’re pleased to inform you that your manuscript has been judged scientifically suitable for publication and will be formally accepted for publication once it meets all outstanding technical requirements.

Kind regards,

Patricia Khashayar

Academic Editor

PLOS ONE

Additional Editor Comments (optional):

Reviewers' comments:

Reviewer's Responses to Questions

**Comments to the Author**

1. Does the manuscript provide a valid rationale for the proposed study, with clearly identified and justified research questions?

Reviewer #1: Yes

Reviewer #2: Yes

2. Is the protocol technically sound and planned in a manner that will lead to a meaningful outcome and allow testing the stated hypotheses?

Reviewer #1: Yes

Reviewer #2: Yes

3. Is the methodology feasible and described in sufficient detail to allow the work to be replicable?

Reviewer #1: Yes

Reviewer #2: Yes

4. Have the authors described where all data underlying the findings will be made available when the study is complete?

Reviewer #1: Yes

Reviewer #2: Yes

5. Is the manuscript presented in an intelligible fashion and written in standard English?

Reviewer #1: Yes

Reviewer #2: Yes

6. Review Comments to the Author

You may also provide optional suggestions and comments to authors that they might find helpful in planning their study.

Reviewer #1: All my concerns are addressed.

The statistical section is acceptable.

Reviewer #2: To the authors,

Thank you for your thoughtful responses to my questions.

I am satisfied that you have adequately addressed all of my concerns regarding the study protocol. I do not have any further questions or comments.

7. PLOS authors have the option to publish the peer review history of their article (what does this mean?). If published, this will include your full peer review and any attached files.

Reviewer #1: No

Reviewer #2: No

---

## [Editor Report · Acceptance letter]

21 Jun 2024

PONE-D-24-12706R1 

PLOS ONE

Dear Dr. Kadam, 

I'm pleased to inform you that your manuscript has been deemed suitable for publication in PLOS ONE. Congratulations! Your manuscript is now being handed over to our production team.

Kind regards, 

on behalf of

Dr. Patricia Khashayar 

Academic Editor

PLOS ONE